# Synthetic data in cancer and cerebrovascular disease research: A novel approach to big data

**Ronda Lun** [1,2,3] *, **Deborah Siegal** [4,5], **Tim Ramsay** [4], **Grant Stotts** [3], **Dar Dowlatshahi** [1,2,3]

**1** School of Epidemiology and Public Health, University of Ottawa, Ottawa, Canada, **2** Clinical Epidemiology Program, Ottawa Hospital Research Institute, Ottawa, Canada, **3** Division of Neurology, Department of Medicine, The Ottawa Hospital, Ottawa, Canada, **4** School of Epidemiology, University of Ottawa, Ottawa, Canada, **5** Division of Hematology, Department of Medicine, The Ottawa Hospital, Ottawa, Canada

* rlun@ucalgary.ca

**Data Availability Statement:** Both of the synthetic datasets are included with the Supporting Information files.

**Funding:** The authors received no specific funding for this work.

## Abstract

### Objectives

Synthetic datasets are artificially manufactured based on real health systems data but do not contain real patient information. We sought to validate the use of synthetic data in stroke and cancer research by conducting a comparison study of cancer patients with ischemic stroke to non-cancer patients with ischemic stroke.

### Design

retrospective cohort study.

### Setting

We used synthetic data generated by MDClone and compared it to its original source data (i.e. real patient data from the Ottawa Hospital Data Warehouse).

### Outcome measures

We compared key differences in demographics, treatment characteristics, length of stay, and costs between cancer patients with ischemic stroke and non-cancer patients with ischemic stroke. We used a binary, multivariable logistic regression model to identify risk factors for recurrent stroke in the cancer population.

### Results

Using synthetic data, we found cancer patients with ischemic stroke had a lower prevalence of hypertension (52.0% in the cancer cohort vs 57.7% in the non-cancer cohort, p<0.0001), and a higher prevalence of chronic obstructive pulmonary disease (COPD: 8.5% vs 4.7%, p<0.0001), prior ischemic stroke (1.7% vs 0.1%, p<0.001), and prior venous thromboembolism (VTE: 8.2% vs 1.5%, p<0.0001). They also had a longer length of stay (8 days [IQR 3–16] vs 6 days [IQR 3–13], p = 0.011), and higher costs associated with their stroke encounters: $11,498 (IQR $4,440 –$20,668) in the cancer cohort vs $8,084 (IQR $3,947 –

**Competing interests:** The authors have declared that no competing interests exist.

$16,706) in the non-cancer cohort (p = 0.0061). A multivariable logistic regression model identified 5 predictors for recurrent ischemic stroke in the cancer cohort using synthetic data; 3 of the same predictors identified using real patient data with similar effect measures. Summary statistics between synthetic and original datasets did not significantly differ, other than slight differences in the distributions of frequencies for numeric data.

## Conclusion

We demonstrated the utility of synthetic data in stroke and cancer research and provided key differences between cancer and non-cancer patients with ischemic stroke. Synthetic data is a powerful tool that can allow researchers to easily explore hypothesis generation, enable data sharing without privacy breaches, and ensure broad access to big data in a rapid, safe, and reliable fashion.

## Introduction

Cancer is the leading cause of mortality in Canada, accounting for approximately 30% of all deaths [1]. In addition to the direct oncologic effects of cancer and cancer-associated treatments, patients with cancer are at an increased risk for thrombotic complications, including ischemic stroke [2–4]. Cancer-associated stroke has been proposed to be a separate entity from stroke in the general population, and identifying key differences in risk factors for development, treatment patterns, and outcomes is crucial in these patients [5]. Due to a lack of randomized controlled trial (RCT)-level evidence in this specific subpopulation, most of what we know about cancer and stroke has come from observational/administrative data. Administrative health data are routinely collected data from interactions with the healthcare system for the purpose of payment, monitoring, planning, priority setting, and evaluation of health service provision [6]. This wealth of data can be used for disease modeling, risk analysis, quality improvement, and much more. However, for the purpose of research/quality improvement, obtaining access to administrative data can be time consuming and significant issues exist surrounding patient confidentiality, allocation of internal resources, delays due to ethics approval, and privacy concerns surrounding data sharing.

Synthetic data is a novel concept in epidemiological cancer and stroke research where patient health data are artificially manufactured based on real health system. For synthetic data to be useful in healthcare research, statistical properties must be nearly identical to original data and could be analyzed as such [7]. The advantages to using synthetic data include reducing barriers to access by bypassing privacy and confidentiality controls, improving the ability to share informative datasets between researchers at different institutions, reducing timing delays in obtaining data, and decreasing the utilization of internal resources, such as information technology (IT) teams. MDClone (Beer Sheva, Israel) is a data synthesis platform that uses computational derivation methods to produce such deidentified, synthetic datasets based on real health systems data. Previous studies using MDClone have demonstrated its utility and efficiency, as the data was shown to maintain similar statistical properties compared to their real data sources. This has proven to be particularly useful when rapid access to big data are required to make informative decisions regarding public health policies and interventions, such as during the COVID-19 pandemic [8, 9]. However, the utility of synthetic data has never been demonstrated in stroke or cancer research.

It is important to establish the utility of synthetic data in cancer and stroke, given the increasing recognition of cancer-associated stroke and the challenges with conducting clinical trials in this high-risk population [5]. Thus, our objectives for this study are: 1) to compare key differences in demographics, acute treatments, hospital length of stay, and costs between cancer patients with ischemic stroke and non-cancer patients with ischemic stroke using synthetic data produced by MDClone, 2) to validate the use of synthetic data in cancer and stroke research by comparing key statistical properties between the synthetic dataset and the source dataset from which the synthetic data originates, and to explore the statistical correlations and properties of both datasets via the building of a multivariable predictive model for recurrent ischemic stroke in cancer patients with stroke.

## Methods

Patients or the public were not involved in the design, or conduct, or reporting, or dissemination plans of our research.

### Standard protocol approvals and patient consents

The Ottawa Hospital is a tertiary care center that is the primary stroke center for a catchment population of 1.6 million in the Eastern Ontario region. We analyzed data through the MDClone platform, which produces synthetic data based on real health system data from the Ottawa Hospital Data Warehouse–a relational database that has been widely used in previous research and contains information that links several hospital information systems, including patient registration, clinical data, case costing, and patient abstract systems. Extensive assessments of data quality were performed during its development and continue to be executed routinely as new data are added [10, 11]. MDClone is an interactive platform that synthesizes existing data from the Ottawa Hospital Data Warehouse and reproduces an imitation dataset that no longer contains data from individual patients, but rather a collection of observations which maintain the statistical properties of the original dataset [7]. Next, real-patient data was obtained from the Ottawa Hospital Data Warehouse for comparison. Patient consent was not required as this was a retrospective chart review with deidentified data. The current study was approved by the Ottawa Hospital Health Science Network Research Ethics Board (Reference number or ID: 20210399-01H). Informed consent from patients was not required as this was a retrospective chart review.

### Patient and public involvement

No patients were involved in the design of this study.

### Inclusion/exclusion criteria

We included all patients meeting inclusion criteria from April 1, 2002 –May 31, 2019. We created two separate cohorts: the first cohort included all patients with a diagnosis of cancer (excluding non-melanoma skin cancer or primary central nervous system [CNS] malignancies) and a diagnosis of ischemic stroke within a 2-year window before and after their cancer diagnosis. The list of cancer subtypes searched for are outlined in S1 Table in S1 File. Patients with non-melanoma skin cancers (i.e. squamous or basal cell carcinoma) were excluded due to their favourable prognoses. Primary CNS malignancies were excluded because of the potential for misdiagnoses of acute vascular events in the context of tumours on neuroimaging [12]. The two-year window was chosen because there appears to be an elevated risk for arterial thromboembolism in patients with cancer up to 24 months after a new diagnosis of cancer; the

cumulative incidence curves for ischemic stroke start to cross over at 24 months [13]. There is also an increased risk for stroke as early as two years before a cancer diagnosis, peaking around the time of diagnosis [14]. Therefore, the two-year period immediately preceding and subsequent to a diagnosis of cancer was broadly chosen to be comprehensive and inclusive of all strokes potentially related to a cancer diagnosis.

The second patient cohort consisting of ischemic stroke patients without a history of cancer was created using administrative coding for "ischemic stroke". All patients with ischemic stroke and no record of any malignancy documented in their chart were included for analysis. Stroke encounters were identified using International Classification of Diseases 10th edition (ICD-10) codes. The list of codes can be found now in S3 Table in S1 File. For Emergency Department visits, encounters were only included if stroke was the "most responsible diagnosis" associated with the visit. For hospitalizations, encounters were included if ischemic stroke was defined as the "primary problem".

## Outcomes

Covariates included for analysis include baseline patient demographics including age, sex, Charlson Comorbidity Index (CCI), presence of vascular risk factors, history of ischemic and hemorrhagic stroke, and history of venous thromboembolism. We also included biochemical information such as blood glucose at time of stroke presentation. Outcomes of interest included the proportion of patients receiving intravenous thrombolysis and the proportion of patients who underwent thrombectomy. We also analyzed the total cost associated with each stroke encounter. Information regarding costs were available for all inpatient encounters from April 2002, and for ED encounters, from April 2011. Direct costs were defined as all expenses in direct functional centers related to patient care, including salaries, supplies, and equipment amortization. Indirect costs included overhead allocation based on the percentage of the activity in the functional center, and consisted of costs not directly related to the patient, such as human resources, finance, health records, administration fees, building maintenance, etc. Mortality was captured from documented deaths in our administrative database. The primary outcome of interest for the cancer cohort was recurrent ischemic stroke at least 14 days after the reference event. This definition was chosen because exploratory analyses conducted using the definition "any recurrent ischemic stroke" demonstrated erroneously high rates of recurrent stroke in both cohorts, likely a reflection of error in administrative coding due to duplicate encounters created for the same presentation.

## Statistical analysis

First, we described the cancer and non-cancer cohorts using descriptive statistics. We compared statistical properties of the cancer cohort generated by MDClone (synthetic cohort) to the cohort obtained from TOH Data repositories (real-patient cohort). Normality of distribution was assessed using histograms; results with a normal distribution were compared using parametric tests for between-group comparisons (i.e. t-test for continuous outcomes and chi-squared test for dichotomous outcomes). Categorical variables were compared using the two-sample t-test. Non-parametric tests were performed for values with non-normal distribution, as appropriate.

We used a binary, multivariable logistic regression model to identify risk factors for recurrent stroke in cancer patients. Variables assessed for inclusion in the model included: sex, age, hypertension, coronary artery disease, chronic obstructive pulmonary disease, diabetes mellitus, dyslipidemia, history of ischemic stroke, atrial fibrillation, venous thromboembolism, and Charlson Comorbidity Index (CCI). Age was an ordinal variable that we recategorized into 5

subgroups: <50, 50–59, 60–69, 70–79, and ≥80. We first performed exploratory univariate analyses to determine associations between candidate variables and the primary outcome, using "any recurrent ischemic stroke" for the purpose of comparing synthetic data to real patient data. We then used a backwards selection, stepwise approach to evaluate the importance of individual covariates for inclusion in the final multivariable logistic regression model using the likelihood ratio test, set at a p value of <0.10 for covariate inclusion. Important collinearity was assessed with variance inflation factor (VIF) values. All statistical analyses were performed using SPSS v27.0, (IBM, Armonk, NY).

## Results

### Objective 1

In total, we included 10,900 synthetic patients from datasets for analysis, with 1,275 subjects in the cancer cohort and 9,625 in the non-cancer cohort. The baseline patient characteristics and demographics are listed in Table 1. There was no difference between the cancer and non-cancer cohorts in the age of presentation at the time of their ischemic stroke: 73.0±12.0 in the cancer cohort vs 73.6±12.6 in the non-cancer cohort (p = 0.18). There was a higher proportion of males in the cancer cohort (55.2% vs 44.7%, p<0.0001). We found no substantial difference in the proportion of patients with diabetes, dyslipidemia, history of hemorrhagic stroke, transient ischemic attack (TIA), atrial fibrillation, or coronary artery disease (Table 1). However, there was a lower prevalence of hypertension (52.0% in the cancer cohort vs 57.7% in the non-cancer cohort, p<0.0001), and a higher prevalence of chronic obstructive pulmonary disease (COPD: 8.5% vs 4.7%, p<0.0001), prior ischemic stroke (1.7% vs 0.1%, p<0.001), and prior venous thromboembolism (VTE: 8.2% vs 1.5%, p<0.0001).

The distribution of cancer types included in the cancer and ischemic stroke cohort are outlined in S2 Table in S1 File. The most frequent cancer diagnosis was lung cancer, which was present in 242/1275 patients (19.0%). The next most frequent cancer type were secondary malignancies, defined as any malignancy that has emerged as a consequence of prior treatment for another primary malignancy [15, 16]. Together with colorectal cancer (152/1275), bladder cancer (95/1275), and immunoproliferative disease (87/1275), these five subtypes of cancer made up 59.5% of the included patients. Of note, there were 6.2 of patients whose primary cancer site was unspecified, and 6.3% of patients that had their cancer type censored by MDClone.

There were similar proportions of patients that received acute treatments for their ischemic stroke between the cancer and non-cancer cohorts: 1.4% of patients in the cancer cohort received IV thrombolysis compared to 1.7% in the non-cancer cohort (p = 0.36). Similarly, 2.0% of patients in the cancer cohort underwent endovascular thrombectomy compared to 2.5% in the non-cancer cohort (p = 0.21). The median length of stay was noted to be longer in the cancer cohort: 8 days (IQR 3–16) vs 6 days (IQR 3–13), (p = 0.011). Overall, costs associated with the stroke encounter were significantly higher for cancer patients compared to non-cancer patients: $11,498 (IQR $4,440 –$20,668) vs $8,084 (IQR $3,947 –$16,706), respectively (p = 0.0061). We also calculated the median and average costs associated with stroke encounters for each year between 2005–2019 and adjusted the costs for inflation based on the 2022 Canadian consumer price index [17]. The average costs associated with encounters in the cancer and stroke cohort was consistently higher compared to encounters in the non-cancer and stroke cohort (S4 Table in S1 File). The rates of recurrent ischemic stroke were similar between the two cohorts: 11.9% in the cancer cohort and 12.1% in the non-cancer cohort (p = 0.86). Lastly, a higher proportion of patients died in the cancer cohort: 36.5% compared to 13.6% in the non-cancer cohort (p<0.0001).

**Table 1. Baseline demographics, treatment characteristics, and outcomes for cancer patients diagnosed with ischemic stroke within a 2-year period compared to patients with ischemic stroke and no history of malignancy.** Data is synthetic and generated from MDClone.

| | Cancer Patients with ischemic stroke | Non-Cancer Patients with ischemic stroke | P-Vaues |
|---|---|---|---|
| **Number of patients** | N = 1275 | N = 9625 | |
| Age: cancer diagnosis (mean ±SD) | 72.9±12.0 | N/A | — |
| Age: stroke diagnosis (mean ±SD) | 73.1±12.0 | 73.6±12.6 | 0.18 |
| Male sex (n, %) | 704 (55.2%) | 4304 (44.7%) | <0.0001 |
| Hypertension | 663 (52.0%) | 5556 (57.7%) | <0.0001 |
| Diabetes | 354 (27.8%) | 2787 (29.0%) | 0.38 |
| Hyperlipidemia | 61 (4.8%) | 584 (6.1%) | 0.068 |
| COPD | 108 (8.5%) | 455 (4.7%) | <0.0001 |
| Ischemic Stroke | 22 (1.7%) | 13 (0.1%) | <0.0001 |
| Hemorrhagic Stroke | 23 (1.8%) | 234 (2.4%) | 0.17 |
| Atrial Fibrillation | 289 (22.7%) | 1945 (20.2%) | 0.041 |
| Coronary Artery Disease | 157 (12.3%) | 1434 (14.9%) | 0.014 |
| Previous Venous Thromboembolism | 105 (8.2%) | 143 (1.5%) | <0.0001 |
| Charlson Comorbidity Index (median, IQR) | 1.0 (0.0–2.0) | 1.0 (1.0–1.0) | 0.97 |
| Initial Glucose (mmol/L) (median, IQR) | 6.4 (5.7–7.8) | 6.8 (5.6–8.6) | 0.022 |
| Received Intravenous tPA | 17 (1.4%) | 162 (1.7%) | 0.36 |
| Received Thrombectomy | 25 (2.0%) | 244 (2.5%) | 0.21 |
| Length of stay (days; median, IQR) | 7.65 (3.11–15.86) | 6.09 (2.81–12.75) | 0.011 |
| Total cost of encounter (CAD; median, IQR) | $11,497.78 (4439.85–20,668.41) | $8084.24 (3947.40–16,706.04) | 0.0061 |
| **Outcomes** | | | |
| Recurrent ischemic stroke after 14 days | 152 (11.9%) | 1163 (12.1%) | 0.86 |
| Timing of recurrent IS after 14d (days; median, IQR) | 25.4 (18.8–48.2) | 26.4 (18.5–64.5) | 0.49 |
| Deaths | 465 (36.5%) | 1313 (13.6%) | <0.0001 |

## Objective 2

MDClone generated a synthetic dataset of 1,275 patients with cancer and ischemic stroke, compared to 1,246 patients from the Ottawa Hospital Data Warehouse. Table 2 summarizes the clinical variables obtained from both the synthetic and original datasets. Most summary statistics were similar between the two datasets, including the proportions of patients with comorbidities, proportions of patients receiving acute treatments, and proportion of patients with mortality or recurrent stroke outcomes. The mean age at cancer diagnosis in the synthetic dataset was 72.9±12.0 years, compared to 72.4±12.1 in the original dataset (p = 0.30); the mean age at stroke diagnosis was similar also (73.0±12.0 vs 72.6±12.0, p = 0.40). Fig 1 displays the histograms of the frequencies of age at cancer diagnosis. While the distributions were relatively normal in both datasets, there were visibly less outliers in the synthetic dataset; the resulting histogram has a slight left-sided skew (Fig 1A). The only other difference noted between the two cohorts was the timing of when recurrent stroke occurred: the synthetic cohort reported a median of 25.4 days (18.8–25.4), while the original cohort found a median of 30.3 days (20.5–76.6), although this was not statistically significant (p = 0.078).

Using the synthetic dataset produced by MDClone, we identified 5 predictors of the primary outcome "recurrent ischemic stroke after the reference event" with a binary logistic regression model. Older age, a history of ischemic stroke, and dyslipidemia were found to be positive predictors of the outcome, while atrial fibrillation and a history of venous thromboembolism were negative predictors (Table 3). Using the original dataset, we identified 3 predictors of recurrent stroke: atrial fibrillation and venous thromboembolism were consistently found

**Table 2. A comparison of the baseline characteristics and outcomes between the synthetic dataset generated from MDClone and real patient dataset for a cohort of stroke patients with a diagnosis of cancer at the Ottawa Hospital.**

| | Cancer patients with ischemic stroke (Synthetic) | Cancer patients with ischemic stroke (Original dataset) | P-Value |
|---|---|---|---|
| Number of patients | N = 1275 | N = 1246 | |
| Age: cancer diagnosis (mean ±SD) | 72.9±12.0 | 72.4±12.1 | 0.30 |
| Age: stroke diagnosis (mean ±SD) | 73.0±12.0 | 72.6±12.0 | 0.40 |
| Male sex | 704 (55.2%) | 688 (55.2%) | 1.00 |
| Hypertension | 663 (52.0%) | 654 (52.5%) | 0.81 |
| Diabetes | 354 (27.8%) | 347 (27.9%) | 0.96 |
| Hyperlipidemia | 61 (4.8%) | 61 (4.9%) | 0.90 |
| COPD | 108 (8.5%) | 105 (8.4%) | 0.97 |
| Ischemic Stroke | 22 (1.7%) | 22 (1.8%) | 0.94 |
| Hemorrhagic Stroke | 23 (1.8%) | 23 (1.9%) | 0.94 |
| Atrial Fibrillation | 289 (22.7%) | 283 (22.7%) | 0.98 |
| Coronary Artery Disease | 157 (12.3%) | 155 (12.4%) | 0.92 |
| Previous Venous Thromboembolism | 105 (8.2%) | 104 (8.4%) | 0.92 |
| Charlson Comorbidity Index (median, IQR) | 1.0 (0.0–2.0) | 1 (0–2) | 0.39 |
| Initial Glucose (mmol/L) (median, IQR) | 6.4 (5.7–7.8) | 6.4 (5.5–8.2) | 0.86 |
| Received Intravenous tPA | 17 (1.4%) | 17 (1.4%) | 0.95 |
| Received Thrombectomy | 25 (2.0%) | 23 (1.8%) | 0.83 |
| Length of stay (days; median, IQR) | 7.7 (3.1–15.9) | 7.6 (3.6–15.9) | 0.88 |
| Total cost of encounter (CAD; median, IQR) | $11,498 ($4,440 –$20,668) | $11,426 ($4,361–21,939) | 0.95 |
| Outcomes | | | |
| Recurrent ischemic stroke after 14 days | 152 (11.9%) | 147 (11.8%) | 0.92 |
| Timing of recurrent IS after 14d (days; median, IQR) | 25.4 (18.8–48.2) | 30.3 (20.5–76.7) | 0.078 |
| Deaths | 465 (36.5%) | 449 (36.0%) | 0.82 |

to be negative predictors of the outcome, with similar adjusted odds ratios compared to synthetic data (aOR 0.67 [95%CI 0.49–0.91] from synthetic data compared to 0.71 [95%CI 0.52–0.97] from original data for atrial fibrillation; aOR 0.51 [95%CI 0.31–0.85] from synthetic data compared to 0.43 [95%CI 0.25–0.72] from original data for venous thromboembolism). Dyslipidemia was similarly identified as a positive predictor of the outcome (aOR 1.89 [95%CI 1.09–3.27] from synthetic data compared to 1.82 [95%CI 1.06–3.13] for original data). There was no significant association between age or prior ischemic stroke with the outcome–in exploratory analyses, the p-values from their likelihood ratio tests were >0.1, and therefore they were not included in the final regression model. The Cox & Snell $R^2$ value for both final logistic regression models were low: 0.026 for the synthetic dataset and 0.017 for the original dataset; suggesting that only approximately 2.6% of the variation in our primary outcome can be attributed to the predictors in our model.

## Discussion

Using synthetic data, we were able to demonstrate that cancer patients with ischemic stroke have a lower prevalence of hypertension, and a higher prevalence of baseline COPD, previous ischemic stroke, and prior venous thromboembolism (VTE) than non-cancer patients. While their rates of receiving acute reperfusion therapies were similar compared to non-cancer patients with ischemic stroke, the cancer and stroke cohort had a significantly longer length of stay and higher costs than non-cancer individuals with stroke. The accuracy of these findings

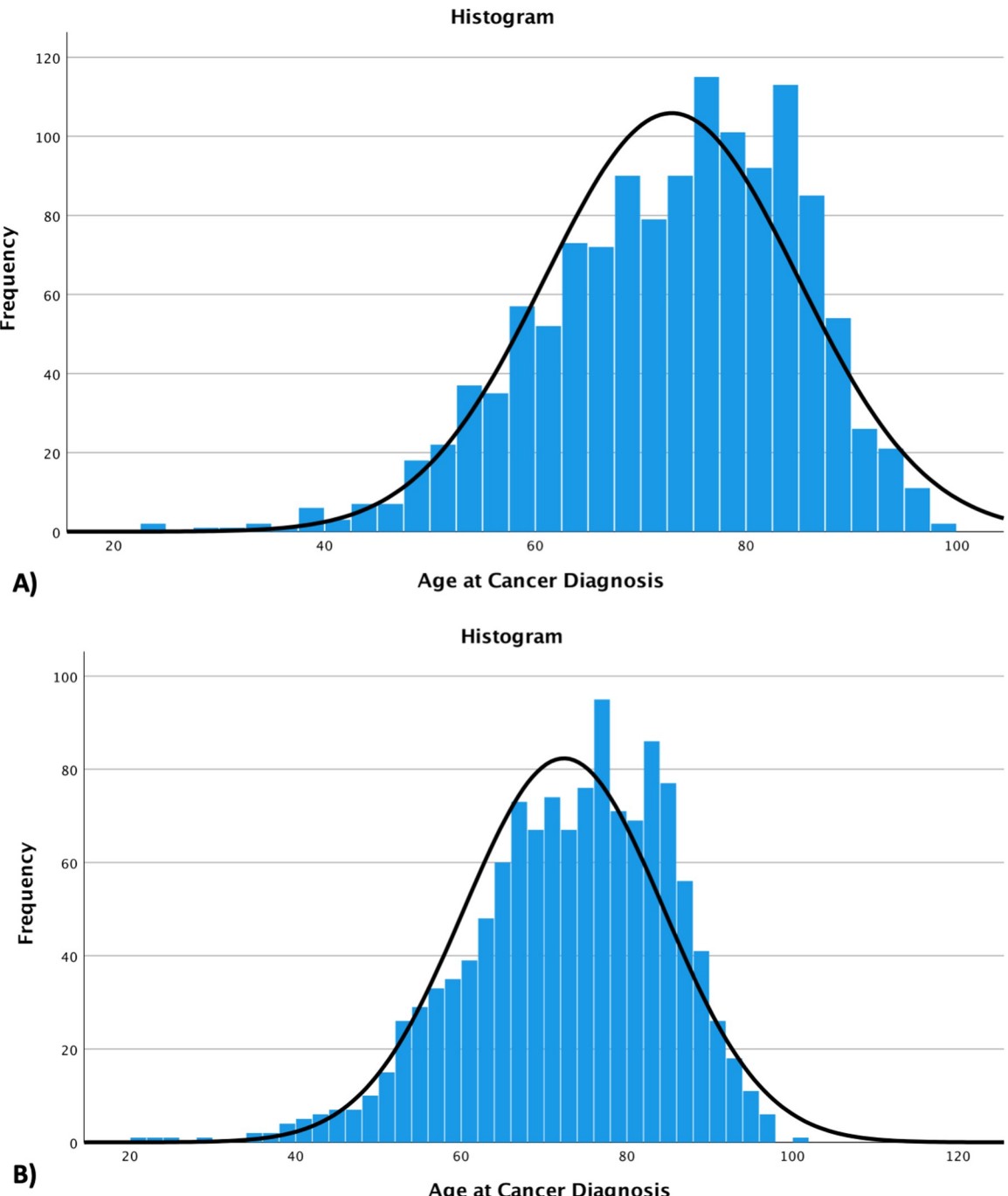

**Fig 1.** Histograms of the distribution of age at cancer diagnosis in the synthetic dataset (A) and the original dataset (B).

was confirmed via a comparison of the synthetic cancer cohort to the original patient data from TOH Data Repositories. No differences were noted in summary statistics between synthetic and real patient datasets other than minor differences in the distribution of synthetic data to real-patient data. Our prediction model for recurrent ischemic stroke in cancer patients with stroke had similarly poor performances using synthetic and original data, with low $R^2$ for both final models. Five predictors were identified using synthetic data, three of which were

**Table 3. Final logistic regression model for the association between covariates and recurrent stroke in the cancer and stroke cohort using synthetic and real patient datasets.** Only covariates meeting requirement for inclusion with likelihood ratio test (p<0.10) are included. Measures of association are reported as adjusted Odds Ratios (aOR) with 95% confidence intervals (95%CI).

| Predictors | Synthetic Dataset | Real Patient Dataset |
|---|---|---|
| Age by decade (reference: <50 years) | 1.16 (95%CI 1.04–1.30) | --- |
| Ischemic stroke | 2.57 (95%CI 1.07–6.18) | --- |
| Atrial fibrillation | 0.67 (95%CI 0.49–0.91) | 0.71 (95%CI 0.52–0.97) |
| Venous thromboembolism | 0.51 (95%CI 0.31–0.85) | 0.43 (95%CI 0.26–0.72) |
| Dyslipidemia | 1.89 (95%CI 1.09–3.27) | 1.82 (95%CI 1.06–3.13) |

also identified using original data and all had similar adjusted odds ratios. Overall, synthetic data produced similar patterns and results compared to the original data, and supports the use of synthetic data in cerebrovascular and cancer health research.

## Objective 1: Stroke in cancer patients vs stroke in non-cancer patients

There is conflicting evidence regarding the prevalence of traditional vascular risk factors in cancer vs non-cancer patients with ischemic stroke, with some studies reporting a higher prevalence in non-cancer patients, and others not finding a difference [18–20]. We believe that these differences may reflect the effect modification of age on the relationship between cancer and stroke and the variations in age across these study populations. In a large cross-sectional study of 86,809 adult patients, we demonstrated that the association between cancer and stroke was only significant in younger adults (defined as <40 years of age), and the association was stronger in those without conventional risk factors such as hypertension [21]. In this current study, the prevalence of hypertension was again found to be lower in cancer patients than non-cancer patients, which may suggest unique relationship between hypertension and stroke in cancer patients specifically. The average age at time of stroke diagnosis was 73 for both the cancer and non-cancer cohorts, with similar proportions of other vascular risk factors reported, likely reflecting the increased prevalence of these conditions with age.

The rates of reperfusion therapy provided in Ontario were reported to be 15% in 2019/2020: 12% of ischemic stroke patients received IV thrombolysis with tissue plasminogen activator (tPA) and 5% underwent endovascular thrombectomy (EVT) [22]. The higher proportion of patients receiving acute reperfusion therapies compared to our study period (i.e. 2001–2019) is likely a reflection of the recently implemented guidelines for endovascular thrombectomy and the extended time window for reperfusion [23–26]. Evidence for thrombectomy did not come out until 2015, when five randomized controlled trials demonstrated overwhelming efficacy of thrombectomy for large vessel occlusions [25]. In the current study, we did not find a difference in the rates of acute reperfusion therapies provided between the cancer and non-cancer cohorts, and this is disparate from previously published studies that found cancer patients are less likely to get reperfusion therapies than non-cancer patients [27, 28]. We believe our results should be interpreted cautiously given the low number of patients receiving reperfusion therapies, particularly in the cancer cohort.

Overall, costs were significantly higher for the cancer cohort compared to the non-cancer cohort; this was true for both direct and indirect costs associated with the stroke encounter. For patients with known cancer, this increased cost may be related to more investigations pursued for the purpose of cancer staging, which may guide conversations around goals of care and prognosis. It is also possible that cancer was diagnosed during the workup for stroke, which may also contribute to higher costs in this cohort. In patients with embolic stroke of

undetermined significance (ESUS), between 5–10% of patients go on to receive a diagnosis of cancer from their workup [29–31]. Cancer patients are also at higher risk for in-hospital complications, including bleeding, development of venous thromboembolism, and infections [32, 33]. Furthermore, those with limited life expectancies related to their cancer diagnosis may not be a candidate for rehabilitation services related to their stroke, which may also contribute to a longer length of stay and higher costs.

## Objective 2: Use of synthetic data

We demonstrated that variables derived from synthetic data were comparable to real-patient data in terms of basic statistical properties, other than slight differences in the distributions of frequencies for numeric data, likely due to MDClone's automatic removal of outliers from the synthetic dataset. In the multivariable model, 3/5 predictors were identified with similar odds ratios but two were not identified in the real patient dataset. Several studies have been published using synthetic data produced by MDClone, suggesting the growing recognition and support for its use amongst the scientific community [8, 34]. We were able to find three previous studies that compared synthetic data produced by MDClone to original data [7, 9, 35]. All three studies found that results derived from synthetic data were representative of real data in terms of basic descriptive statistical properties. For variables with large patient numbers, there were highly accurate and strongly consistent results observed compared to original data, but in the context of large missingness of data or small patient numbers, the results of synthetic data were less reliable [7, 35]. One study found that for smaller population studies that evaluated confounders and effect modifiers in multivariable regression models, clear trends were still correctly observed with synthetic data, although the predictions were of moderate accuracy compared to original data [35]. This is in line with the findings from our current study–we believe that the reason previous ischemic stroke was found to be a predictor with synthetic data but not the real dataset is related to the low proportion of patients with this diagnosis: only 22 patients were identified to have a history of ischemic stroke in both the synthetic and original datasets. The high missingness of data would render this covariate unreliable, which is further supported by the wide confidence interval for its adjusted odds ratio (aOR 2.57, 95%CI 1.07–6.18). The other covariate that was identified with the synthetic dataset but not original was age categorized by decades, which may be related to skewing of data related to the automatic removal of outliers by MDClone (Fig 1) and the proximity of the lower 95% CI to 1. A previous study also found that low sample size, highly irregular distributions, and high sparsity of data can all affect the data synthesis process and the interpretability of synthetic data [7].

We believe that MDClone is a useful adjunctive tool for clinical research using administrative databases. The advantages to this synthetic data platform are particularly evident during the early stages of a research inquiry, as it allows the investigator to easily explore project feasibility. This includes being able to predict the scope of a study population, generating hypotheses, and exploring relationships between covariates before applying for ethics approvals. Its deidentified nature also allows for data sharing between institutions, without concern for patient privacy breaches. However, this platform is not without limitations. We were able to demonstrate that while most properties of descriptive statistics were maintained between original and synthetic data, significant proportions of missing data and automatic elimination of outliers may affect results. There were also some automatically calculated variables (i.e. the Charlson Comorbidity Index), where the individual components of the score could not be individually analyzed. Categorical variables may be subject to censoring if an indirect request for numeric values cannot be made, rendering statistical analysis challenging due to a high proportion of missing data. Therefore, the utility of MDClone may be during the early,

hypothesis-generating stage of a research project; a robust and complete dataset with reliable variable derivation methods may require access to original data. Lastly, the accuracy of synthetic data is only as reliable as the administrative database it derives its information from. Pitfalls related to the use of administrative data for healthcare research are beyond the scope of this report, but common criticisms pertaining its use include misclassification bias, coding inaccuracies, lack of reliable, timed follow-up, and lack of quality control of data [6, 36, 37].

## Conclusion

Using synthetic data, we were able to demonstrate that cancer patients with ischemic stroke differ from non-cancer patients with ischemic stroke in terms of baseline demographics, longer average lengths of stay, and higher average costs associated with their stroke encounter. These results were confirmed by comparing synthetic data to original administrative data. Synthetic data is a powerful tool that can allow researchers to easily explore hypothesis generation, enable data sharing without privacy breaches, and ensure broad access to big data in a rapid, safe, and reliable fashion.

## Supporting information

**S1 File.**
(PDF)

**S1 Data. Synthetic deidentified study dataset–cancer patients.**
(XLSX)

**S2 Data. Synthetic deidentified study dataset–non-cancer patients.**
(XLSX)

## Author Contributions

**Conceptualization:** Ronda Lun, Dar Dowlatshahi.

**Data curation:** Ronda Lun.

**Formal analysis:** Ronda Lun, Tim Ramsay.

**Investigation:** Ronda Lun.

**Methodology:** Ronda Lun, Dar Dowlatshahi.

**Resources:** Grant Stotts.

**Software:** Ronda Lun.

**Supervision:** Ronda Lun, Deborah Siegal, Tim Ramsay, Grant Stotts, Dar Dowlatshahi.

**Writing – original draft:** Ronda Lun.

**Writing – review & editing:** Ronda Lun, Deborah Siegal, Tim Ramsay, Grant Stotts, Dar Dowlatshahi.

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
