## [Decision Letter · Decision Letter 0]

24 May 2023

PONE-D-23-05410Synthetic Data in Cancer and Cerebrovascular Disease Research: A Novel Approach to Big Data Synthetic Data in Cancer/Stroke ResearchPLOS ONE

Dear Dr. Lun,

Thank you for submitting your manuscript to PLOS ONE. After careful consideration, we feel that it has merit but does not fully meet PLOS ONE’s publication criteria as it currently stands. Therefore, we invite you to submit a revised version of the manuscript that addresses the points raised during the review process.

We look forward to receiving your revised manuscript.

Kind regards,

Rittal Mehta

Academic Editor

PLOS ONE

Journal Requirements:

6. We note that you have referenced (Lun et al, unpublished data, manuscript embargo) which has currently not yet been accepted for publication. Please remove this from your References and amend this to state in the body of your manuscript: (ie “Bewick et al. [Unpublished]”) as detailed online in our guide for authors

Additional Editor Comments:

In the current manuscript, the authors’ objectives were two-fold: the first was to compare key differences in demographics, acute treatments, hospital length of stay, and costs between cancer patients with ischemic stroke and non-cancer patients with ischemic stroke using synthetic data produced by MDClone, and 2) to validate the use of synthetic data in cancer and stroke research by comparing key statistical properties between the synthetic dataset and the source dataset from which the synthetic data originates.

The reviewer finds the use of MDClone fascinating and recognizes its value in terms of doing feasibility analysis or hypothesis generation. However, the reviewer Is concerned regarding the conclusion regarding objective 1 absent any multivariable analysis. The reviewer also noticed there were no p-values in any of the tables. The authors are advised to perform multivariable analyses to quantify the association of presence of cancer with outcomes of interest among patients with ischemic stroke; and to add p-values to both the tables in their univariate analysis

To analyze the data and answer questions mentioned in the aim of the manuscript, appropriate bivariate and multivariable analyses needs to be performed.

Reviewers' comments:

Reviewer's Responses to Questions

**Comments to the Author**

1. Is the manuscript technically sound, and do the data support the conclusions?

Reviewer #1: Yes

2. Has the statistical analysis been performed appropriately and rigorously? 

Reviewer #1: No

3. Have the authors made all data underlying the findings in their manuscript fully available?

Reviewer #1: No

4. Is the manuscript presented in an intelligible fashion and written in standard English?

Reviewer #1: Yes

5. Review Comments to the Author

Reviewer #1: In the current manuscript, the authors’ objectives were two-fold: the first was to compare key differences in demographics, acute treatments, hospital length of stay, and costs between cancer patients with ischemic stroke and non-cancer patients with ischemic stroke using synthetic data produced by MDClone, and 2) to validate the use of synthetic data in cancer and stroke research by comparing key statistical properties between the synthetic dataset and the source dataset from which the synthetic data originates.

The reviewer finds the use of MDClone fascinating and recognizes its value in terms of doing feasibility analysis or hypothesis generation. However, the reviewer Is concerned regarding the conclusion regarding objective 1 absent any multivariable analysis. The reviewer also noticed there were no p-values in any of the tables. The authors are advised to perform multivariable analyses to quantify the association of presence of cancer with outcomes of interest among patients with ischemic stroke; and to add p-values to both the tables in their univariate analysis.

6. PLOS authors have the option to publish the peer review history of their article (what does this mean?). If published, this will include your full peer review and any attached files.

Reviewer #1: No

---

## [Author Response · Author response to Decision Letter 0]

14 Jun 2023

Response to Reviewers – PLOS ONE 

Journal Requirements:

We have included both of the synthetic datasets for patients with cancer and a history of ischemic stroke as well as patients with ischemic stroke that have no history of cancer. 

6. We note that you have referenced (Lun et al, unpublished data, manuscript embargo) which has currently not yet been accepted for publication. Please remove this from your References and amend this to state in the body of your manuscript: (ie “Bewick et al. [Unpublished]”) as detailed online in our guide for authors

These have all been corrected, thank you. 

 

Reviewer #1: In the current manuscript, the authors’ objectives were two-fold: the first was to compare key differences in demographics, acute treatments, hospital length of stay, and costs between cancer patients with ischemic stroke and non-cancer patients with ischemic stroke using synthetic data produced by MDClone, and 2) to validate the use of synthetic data in cancer and stroke research by comparing key statistical properties between the synthetic dataset and the source dataset from which the synthetic data originates.

The reviewer finds the use of MDClone fascinating and recognizes its value in terms of doing feasibility analysis or hypothesis generation. However, the reviewer Is concerned regarding the conclusion regarding objective 1 absent any multivariable analysis. The reviewer also noticed there were no p-values in any of the tables. The authors are advised to perform multivariable analyses to quantify the association of presence of cancer with outcomes of interest among patients with ischemic stroke; and to add p-values to both the tables in their univariate analysis.

Thank you for your review. We have added p-values to Tables 1 and 2 and described in our Methods section the statistical tests that were utilized to obtain the p-values. 

We have also performed a multivariable analysis quantifying the association between cancer with the outcome of recurrent ischemic stroke: 

Results: 

Using the synthetic dataset produced by MDClone, we identified 5 predictors of the primary outcome “recurrent ischemic stroke after the reference event” with a binary logistic regression model. Older age, a history of ischemic stroke, and dyslipidemia were found to be positive predictors of the outcome, while atrial fibrillation and a history of venous thromboembolism were negative predictors (Table 3). Using the original dataset, we identified 3 predictors of recurrent stroke: atrial fibrillation and venous thromboembolism were consistently found to be negative predictors of the outcome, with similar adjusted odds ratios compared to synthetic data (aOR 0.67 [95%CI 0.49 – 0.91] from synthetic data compared to 0.71 [95%CI 0.52 – 0.97] from original data for atrial fibrillation; aOR 0.51 [95%CI 0.31 – 0.85] from synthetic data compared to 0.43 [95%CI 0.25 – 0.72] from original data for venous thromboembolism). Dyslipidemia was similarly identified as a positive predictor of the outcome (aOR 1.89 [95%CI 1.09 – 3.27] from synthetic data compared to 1.82 [95%CI 1.06 – 3.13] for original data). There was no significant association between age or prior ischemic stroke with the outcome – in exploratory analyses, the p-values from their likelihood ratio tests were >0.1, and therefore they were not included in the final regression model. The Cox & Snell R2 value for both final logistic regression models were low: 0.026 for the synthetic dataset and 0.017 for the original dataset; suggesting that only approximately 2.6% of the variation in our primary outcome can be attributed to the predictors in our model.

Discussion: 

In the multivariable model, 3/5 predictors were identified with similar odds ratios but two were not identified in the real patient dataset. Several studies have been published using synthetic data produced by MDClone, suggesting the growing recognition and support for its use amongst the scientific community.8,33 We were able to find three previous studies that compared synthetic data produced by MDClone to original data.7,9,34 All three studies found that results derived from synthetic data were representative of real data in terms of basic descriptive statistical properties. For variables with large patient numbers, there were highly accurate and strongly consistent results observed compared to original data, but in the context of large missingness of data or small patient numbers, the results of synthetic data were less reliable.7,34 One study found that for smaller population studies that evaluated confounders and effect modifiers in multivariable regression models, clear trends were still correctly observed with synthetic data, although the predictions were of moderate accuracy compared to original data.7 This is in line with the findings from our current study – we believe that the reason previous ischemic stroke was found to be a predictor with synthetic data but not the real dataset is related to the low proportion of patients with this diagnosis: only 22 patients were identified to have a history of ischemic stroke in both the synthetic and original datasets. The high missingness of data would render this covariate unreliable, which is further supported by the wide confidence interval for its adjusted odds ratio (aOR 2.57, 95%CI 1.07 – 6.18). The other covariate that was identified with the synthetic dataset but not original was age categorized by decades, which may be related to skewing of data related to the automatic removal of outliers by MDClone (Figure 1) and the proximity of the lower 95% CI to 1. A previous study also found that low sample size, highly irregular distributions, and high sparsity of data can all affect the data synthesis process and the interpretability of synthetic data.3

---

## [Decision Letter · Decision Letter 1]

11 Sep 2023

PONE-D-23-05410R1Synthetic data in cancer and cerebrovascular disease research: a novel approach to big data Synthetic data in cancer/stroke researchPLOS ONE

Dear Dr. Lun,

Thank you for submitting your manuscript to PLOS ONE. After careful consideration, we feel that it has merit but does not fully meet PLOS ONE’s publication criteria as it currently stands. Therefore, we invite you to submit a revised version of the manuscript that addresses the points raised during the review process. The manuscript was evolved significantly after last review, but there are still queries to be answered, as reported by Reviewer #2. 

We look forward to receiving your revised manuscript.

Kind regards,

Luiz Sérgio Fernandes de Carvalho, PhD, MSc, MD

Academic Editor

PLOS ONE

Journal Requirements:

Reviewers' comments:

Reviewer's Responses to Questions

**Comments to the Author**

1. If the authors have adequately addressed your comments raised in a previous round of review and you feel that this manuscript is now acceptable for publication, you may indicate that here to bypass the “Comments to the Author” section, enter your conflict of interest statement in the “Confidential to Editor” section, and submit your "Accept" recommendation.

Reviewer #1: All comments have been addressed

Reviewer #2: (No Response)

2. Is the manuscript technically sound, and do the data support the conclusions?

Reviewer #1: Yes

Reviewer #2: Yes

3. Has the statistical analysis been performed appropriately and rigorously? 

Reviewer #1: Yes

Reviewer #2: Yes

4. Have the authors made all data underlying the findings in their manuscript fully available?

Reviewer #1: Yes

Reviewer #2: Yes

5. Is the manuscript presented in an intelligible fashion and written in standard English?

Reviewer #1: Yes

Reviewer #2: Yes

6. Review Comments to the Author

Reviewer #1: The authors have addressed the comments from the reviewer and may now be accepted. The authors are to be commended for this manuscrip.t

Reviewer #2: Thank you for the opportunity to review. This is truly novel and interesting manuscript.

Minor comments and questions to the authors:

1. CCI in outcome analyses: since you're investigating cancer patients and CCI contains cancer variable, have you adjusted CCI to remove cancer and re-calculate CCI?

2. Looking at your original dataset time horizon (2002-2019), which is very long, have you considered adjusting costs for inflation in your analyses to make comparisons valid? Current cost estimates presented in the manuscript should be adjusted for inflation.

3. How was stroke encounter defined, and how would the audience know whether the cost of an encounter can truly be associated with stroke? Also, how were indirect costs calculated?

Thank you!

7. PLOS authors have the option to publish the peer review history of their article (what does this mean?). If published, this will include your full peer review and any attached files.

Reviewer #1: **Yes: **Mohammed Ali Alvi

Reviewer #2: **Yes: **Jan Sieluk

---

## [Author Response · Author response to Decision Letter 1]

16 Oct 2023

Reviewer #1: The authors have addressed the comments from the reviewer and may now be accepted. The authors are to be commended for this manuscrip.t

Thank you for taking the time to review our manuscript. 

Reviewer #2: Thank you for the opportunity to review. This is truly novel and interesting manuscript.

Minor comments and questions to the authors:

1. CCI in outcome analyses: since you're investigating cancer patients and CCI contains cancer variable, have you adjusted CCI to remove cancer and re-calculate CCI?

1. Thank you for this suggestion. The Charlson Comorbidity Index was automatically calculated by MDClone and therefore the cancer variable was not removed and re-calculated. This limitation has been added to the discussion section of our manuscript: “There were also some automatically calculated variables (i.e. the Charlson Comorbidity Index), where the individual components of the score could not be individually analyzed.”

2. Looking at your original dataset time horizon (2002-2019), which is very long, have you considered adjusting costs for inflation in your analyses to make comparisons valid? Current cost estimates presented in the manuscript should be adjusted for inflation.?

Thank you for this suggestion. We have broken down the median cost associated with encounters by year and adjusted each year’s median cost for inflation based on the 2022 Canadian consumer price index (Statistics Canada. Table 18-10-0005-01 Consumer Price Index, annual average, not seasonally adjusted). We have presented this data in the supplemental materials document S4 table. Pooling of the median values from each year was not undertaken as medians are not amenable to further mathematic calculations. However, we have also calculated the mean cost associated with encounters from each year and presented the original mean by year as well as the inflation-adjusted cost (adjusted for 2022), and presented the total mean sum for the cancer cohort (S4 Table A) and the non-cancer cohort (S4 Table B). The original mean for the cancer cohort was $15,314.50 and the inflation-adjusted cost was $20,686.29. Comparatively the original mean for the non-cancer cohort was $14,410.78 and after adjusting for inflation, the 2022 equivalent would be $17,295.31. Our conclusions are therefore similar – that the cancer cohort has higher costs associated with their encounter compared to the non-cancer cohort. We did not present these inflation-adjusted means in the main text of our manuscript because cost was not a normally distributed value and therefore presentation of the original median values would be more statistically correct. We have also added this to the text of our main manuscript: “We also calculated the median and average costs associated with stroke encounters for each year between 2005 – 2019 and adjusted the costs for inflation based on the 2022 Canadian consumer price index.28 The average costs associated with encounters in the cancer and stroke cohort was consistently higher compared to encounters in the non-cancer and stroke cohort (Supplemental Materials S4 Table).” 

We hope this is satisfactory to the reviewer. 

3. How was stroke encounter defined, and how would the audience know whether the cost of an encounter can truly be associated with stroke? Also, how were indirect costs calculated

Stroke encounters were identified using International Classification of Diseases 10th edition (ICD-10) codes. The list of codes can be found now in Supplemental Materials Table S3. For Emergency Department visits, encounters were only included if stroke was the “most responsible diagnosis” associated with the visit. For hospitalizations, encounters were included if ischemic stroke was defined as the “primary problem”. 

Information regarding costs were available for all inpatient encounters from April 2002, and for ED encounters, from April 2011. Direct costs were defined as all expenses in direct functional centers related to patient care, including salaries, supplies, and equipment amortization. Indirect costs included overhead allocation based on the percentage of the activity in the functional center, and consisted of costs not directly related to the patient, such as human resources, finance, health records, administration fees, building maintenance, etc. 

The above statements have been added to our revised manuscript.

---

## [Decision Letter · Decision Letter 2]

4 Dec 2023

Synthetic data in cancer and cerebrovascular disease research: a novel approach to big data Synthetic data in cancer/stroke research

PONE-D-23-05410R2

Dear Dr. Lun,

We’re pleased to inform you that your manuscript has been judged scientifically suitable for publication and will be formally accepted for publication once it meets all outstanding technical requirements.

Kind regards,

Luiz Sérgio Fernandes de Carvalho, PhD, MSc, MD

Academic Editor

PLOS ONE

Additional Editor Comments (optional):

Reviewers' comments:

Reviewer's Responses to Questions

**Comments to the Author**

1. If the authors have adequately addressed your comments raised in a previous round of review and you feel that this manuscript is now acceptable for publication, you may indicate that here to bypass the “Comments to the Author” section, enter your conflict of interest statement in the “Confidential to Editor” section, and submit your "Accept" recommendation.

Reviewer #1: All comments have been addressed

2. Is the manuscript technically sound, and do the data support the conclusions?

Reviewer #1: Yes

3. Has the statistical analysis been performed appropriately and rigorously? 

Reviewer #1: Yes

4. Have the authors made all data underlying the findings in their manuscript fully available?

Reviewer #1: Yes

5. Is the manuscript presented in an intelligible fashion and written in standard English?

Reviewer #1: Yes

6. Review Comments to the Author

Reviewer #1: I have already reviewed the revised version before. The authors have addressed some more comments for another reviewer who recommended some changes to the revised version. Congrats again!

7. PLOS authors have the option to publish the peer review history of their article (what does this mean?). If published, this will include your full peer review and any attached files.

Reviewer #1: **Yes: **Mohammed Ali Alvi

---

## [Editor Report · Acceptance letter]

25 Jan 2024

PONE-D-23-05410R2 

PLOS ONE

Dear Dr. Lun, 

I'm pleased to inform you that your manuscript has been deemed suitable for publication in PLOS ONE. Congratulations! Your manuscript is now being handed over to our production team.

Kind regards, 

on behalf of

Dr. Luiz Sérgio Fernandes de Carvalho 

Academic Editor

PLOS ONE